# Striated Preferentially Expressed Protein Kinase (SPEG) in Muscle Development, Function, and Disease

**DOI:** 10.3390/ijms22115732

**Published:** 2021-05-27

**Authors:** Shiyu Luo, Samantha M. Rosen, Qifei Li, Pankaj B. Agrawal

**Affiliations:** 1Division of Genetics and Genomics, Boston Children’s Hospital and Harvard Medical School, Boston, MA 02115, USA; SHIYU.LUO@childrens.harvard.edu (S.L.); Samantha.Rosen@childrens.harvard.edu (S.M.R.); Qifei.Li@childrens.harvard.edu (Q.L.); 2Division of Newborn Medicine, Boston Children’s Hospital and Harvard Medical School, Boston, MA 02115, USA; 3The Manton Center for Orphan Disease Research, Boston Children’s Hospital and Harvard Medical School, Boston, MA 02115, USA

**Keywords:** striated preferentially expressed gene, centronuclear myopathy, cardiomyopathy, muscle regeneration, satellite cells, triad, sarcomere, excitation-contraction coupling

## Abstract

Mutations in striated preferentially expressed protein kinase (SPEG), a member of the myosin light chain kinase protein family, are associated with centronuclear myopathy (CNM), cardiomyopathy, or a combination of both. Burgeoning evidence suggests that SPEG plays critical roles in the development, maintenance, and function of skeletal and cardiac muscles. Here we review the genotype-phenotype relationships and the molecular mechanisms of SPEG-related diseases. This review will focus on the progress made toward characterizing SPEG and its interacting partners, and its multifaceted functions in muscle regeneration, triad development and maintenance, and excitation-contraction coupling. We will also discuss future directions that are yet to be investigated including understanding of its tissue-specific roles, finding additional interacting proteins and their relationships. Understanding the basic mechanisms by which SPEG regulates muscle development and function will provide critical insights into these essential processes and help identify therapeutic targets in SPEG-related disorders.

## 1. Introduction

Centronuclear myopathies (CNMs) are a group of congenital myopathies characterized by clinical features of muscle weakness, increased central nuclei, and genetic heterogeneity [1]. Tremendous progress in the field of gene discovery has led to remarkable insights into the underlying genetics of these disorders. The most common forms of CNMs have been attributed to X-linked recessive mutations in the *MTM1* gene encoding myotubularin, autosomal dominant mutations in the *DNM2* gene encoding dynamin 2 and the *BIN1* gene encoding bridging integrator 1, and autosomal recessive mutations in *BIN1*, the *RYR1* gene encoding ryanodine receptor 1, the *TTN* gene encoding titin, and the *SPEG* gene encoding striated preferentially expressed gene [2,3,4,5,6,7,8]. CNMs linked to MTM1, DNM2, and BIN1 present with highly similar pathogenesis, and most translational research has focused on these forms over the past decades [9,10,11,12,13]. Recent studies have indicated SPEG, a member of the myosin light chain kinase (MLCK) protein family, in several facets of muscle development and function [14,15,16,17,18,19,20,21,22,23]. This review will give an overview of current advances in understanding the pathogenic mechanisms of *SPEG*-related diseases, with an emphasis on emerging clinical phenotypes and genotype-phenotype relationships, SPEG and its interacting proteins, and its potential roles in muscle regeneration, triad maintenance and excitation–contraction (E-C) coupling. We also summarize the unresolved questions and future lines of enquiry.

## 2. Classification, Evolution and Structure of SPEG

SPEG belongs to the MLCK protein family, a branch of the calcium (Ca^2+^)-calmodulin-regulated kinase superfamily functionally associated with myocyte function and the development and regulation of the actin-based cytoskeleton [24,25,26]. Beyond the classical MLCK proteins (MLCK-1, -2, -3, -4) [27], this family includes titin and related invertebrate proteins projectin and stretchin-MLCK [28], trio subfamily consisting of trio and unc-73 in invertebrates and trio and kalirin in vertebrates [29], and the Unc89 subfamily comprised of unc-89 in invertebrates and obscurin, obscurin-like 1 (OBSL1), and SPEG in vertebrates [30,31]. The hallmark protein motifs of the MLCK family include the immunoglobulin (Ig), fibronectin Type-III (FnIII), and serine/threonine kinase (SK) domains. The Ig and FnIII domains interact with other proteins, anchoring the kinase to specific intracellular sites and enabling interactions with its substrates. They can also support a physical separation between functional domains of the protein to allow spatially distinct signaling interactions within a large protein assembly. In striated muscle, MLCKs primarily regulate muscle contraction and the actomyosin cytoskeleton in response to cytosolic alterations in Ca^2+^ [32].

The Unc89 kinases are structurally defined by a large amino-terminal array of Ig repeats, FnIII domains, a RhoGEF guanine nucleotide exchange factor domain, and tandem serine/threonine kinase domains at the carboxy-terminus [24,25]. Deviating from this composition is SPEG which features a truncated N-terminal Ig array and lacks a RhoGEF domain, and OBSL1 which does not contain the RhoGEF and kinase domains [31]. Additionally, all members of the Unc89 kinase subfamily are extensively alternatively spliced, in some cases giving rise to isoforms that lack one or both of the catalytic regions [30]. These kinases are predominantly expressed in skeletal muscle and are the only known members of the MLCK family containing dual kinase motifs except OBSL1 [33]. The evolutionary origins of the Unc89 subfamily can be traced back to the invertebrate kinase unc-89, a giant multi-domain protein with a critical role in myofibril assembly [29]. Phylogenetic analysis of the kinase domains revealed a likely orthologous relationship with the vertebrate protein obscurin [25]. The *OBSCN* gene encodes 117 exons that are alternatively spliced to produce various isoforms of the obscurin protein family, which range in size from 50 to 870 kDa [34]. SPEG, lacking an invertebrate orthologue and demonstrating the highest degree of sequence similarity with obscurin, is theorized to have arisen from obscurin through a gene duplication event [25]. SPEG shares unique homology with obscurin as both contain two tandemly arranged kinase domains (SK1 and SK2) [35,36]. They also contain Ig and FnIII domains that typify this protein family.

The arrangement of functional domains across MLCK family members is highly conserved, characterized by a dual kinase motif immediately preceded by an Ig domain and flanking sequential Ig and FnIII domains. However, the primary amino acid sequence immediately upstream of the carboxy terminal kinase domain is significantly longer in SPEG compared with related proteins, effectively increasing the physical distance between the carboxy terminal kinase and the other functional domains. This distance is highly conserved across species for each kinase, indicating that the spatial relationship between kinase domains is critical to their catalytic activities. One of the differences between SPEG and obscurin is that the exon preceding SPEG SK2 has an additional 525 base pairs, which increases the distance between SPEG SK2 and the preceding Ig/FnIII domain, suggestive of potential differences in their binding partners [25]. However, it is unknown how this difference in the primary sequence affects the three-dimensional structure and true distance between the kinase domains. Surprisingly, while the distances between the kinase domains vary between obscurin and SPEG, the distances between the Ig domains upstream of each amino terminal kinase domain are similar. Thus, whereas consistent distances between Ig and FnIII domains indicate that SPEG and obscurin maintain highly conserved and potentially overlapping localization sites [25,37], distinct spatial properties of the dual kinase motifs suggest unique phosphorylation targets [25]. Obscurin is thought to link the contractile apparatus in muscle cells with longitudinal sarcoplasmic reticulum (SR) [38,39], whereas SPEG localizes in a double line of skeletal muscle, in alignment with the terminal cisternae of the SR and plays a critical role in maintaining triad structure and calcium handling [15].

## 3. SPEG Isoforms and Expression

The *SPEG* locus is subject to extensive alternative splicing [24,40], giving rise to four major isoforms (Figure 1) which were initially characterized by Hsieh and colleagues [24] in several murine tissues. Aortic preferentially expressed protein 1 (Apeg1) is a 1.4-kb transcript encoding a 12.7 kDa protein expressed predominantly in vascular smooth muscle cells (VSMCs). Brain preferentially expressed protein (Bpeg) is a 4-kb isoform expressed exclusively in brain and aorta [24]. Of note, neither Apeg1 nor Bpeg contain the kinase domains characteristic of Unc89 kinases. Spegα and Spegβ are 9-kb and 11-kb transcripts encoding 250 kDa and 355 kDa proteins, respectively, both of which are predominantly expressed in skeletal and cardiac muscle [24].

In addition to tissue specificity, expression of the different *SPEG* isoforms is developmentally regulated. In vitro studies in skeletal muscle found minimal expression of SPEGα and SPEGβ during rapid growth of C2C12 myoblasts, though differentiation of myoblasts into myotubes correlated with a distinct rise in SPEGα expression that persisted throughout development; SPEGβ was upregulated slightly later and gained slight predominance by adulthood [24]. Additionally, in vivo cardiac studies in mice have discerned a dramatic shift in isoform predominance from Apeg1 in the embryonic heart to Spegα and Spegβ in the immediate postnatal period, closely correlating with the timing of neonatal cardiomyocyte maturation [41]. In contrast, others have reported the presence of all four isoform transcripts and both Spegα and Spegβ proteins in fetal heart [36], though their induction is consistently correlated with muscle cell differentiation. Together, these findings indicate that the various *SPEG* isoforms serve as sensitive markers for the early stages of muscle development.

Chromosomal positioning may also affect the expression of the *SPEG* complex locus due to its proximity to the muscle-specific desmin gene (*DES*) locus control region (LCR), a transcriptional regulatory element hypothesized to affect the tissue-specific expression of both *SPEG* and the adjacent *DES* in muscle [40]. Although this relationship has not yet been experimentally verified, functional association between SPEG and desmin in muscle is illustrated by their co-localization at the Z line, overlapping molecular interactomes, coordinate expression patterns, and related sarcomeric functions [8,22,24]. Together, these findings support the notion of a common regulatory factor at the transcriptional level. Elucidating the complex relationship between these genetic elements will shed light on the mechanisms underlying the development and maintenance of the sarcomere.

## 4. *SPEG* Mutations in Myopathies and Cardiomyopathies

Recessive mutations in *SPEG* (MIM 615950) have been linked to a clinically heterogeneous condition known as CNM with or without dilated cardiomyopathy (DCM) [8,42,43,44]. CNMs are a subtype of congenital myopathies (CM) clinically characterized by hypotonia and muscle dysfunction ranging in severity from mild delays in motor milestones to fatal weakness of respiratory organs [10]. Underlying these clinical phenotypes, cellular and molecular hallmarks of CNM include increased central nucleation of myofibers and variation in myofibers size, and disruptions in triadic structure and E-C coupling [10,44,45]. To date, 20 patients have been identified with *SPEG* mutations [23,44,46,47,48,49]. Figure 1 shows the location and mutation type of all reported *SPEG* variants. Two patients lacking centralized myonuclei were diagnosed with non-specific CM [44,50]. Additional clinical features frequently observed in *SPEG*-related patients include respiratory issues, ophthalmoplegia, and scoliosis [8,43,44,50,51]. Further, reduced protein levels of SPEG have also been found as potential contributors to the development of heart failure (HF) and atrial fibrillation [14,21].

Mutations affecting both *SPEG* isoforms are associated with more severe clinical and molecular phenotypes than those that spare *SPEG*α, the shorter isoform lacking amino acids 1-854 [8]. Of the eight patients shown to carry mutations affecting both *SPEG*α and *SPEG*β, all demonstrated cardiac dysfunction, five experienced early death, and one requires constant mechanical ventilation due to severe respiratory problems [8,43,44,46,51]. Mutations affecting only *SPEG*β are associated with a milder phenotype without cardiac involvement [43,44,50], indicating that SPEGα is critical for both skeletal and cardiac function and may be able to partially compensate for SPEGβ loss-of-function.

Five additional patients from a single family showing isolated DCM without myopathy was recently described by Levitas et al. [47] carrying homozygous missense mutation (c.5038G>A, p.Glu1680Lys) in the serine/threonine protein kinase (SK)-1 domain of SPEG. Functional studies in induced pluripotent stem cell-derived cardiomyocytes showed aberrant calcium homeostasis, impaired contractility, and sarcomeric disorganization [47]. We recently identified three patients with a homozygous in-frame deletion (c.9028_9030delGAG, p.Glu3010del) in the SK-2 domain of SPEG [49]. Structure-based alignment indicated that both residues (glutamate) are highly conserved and may function as nucleotide binding sites [52]. The SK-1 domain of SPEG has been shown to phosphorylate junctophilin-2 (JPH2) in cardiac muscle, while SK-2 phosphorylates sarco-endoplasmic reticulum ATPase-2a (SERCA2a) [14,19]. Further investigations are needed to understand how these glutamate residue mutations may specifically affect the function of SPEG in the cardiac muscle.

## 5. Role of SPEG in Muscle Function and Regeneration

Skeletal muscle contraction requires the conversion of electrical stimuli into a mechanical action through a process known as E-C coupling [53]. This process occurs within tripartite structures located at the Z-line of the sarcomere known as triads, consisting of a transverse tubule (T-tubule) invagination of the sarcolemma flanked by two terminal cisternae of the SR. Voltage-sensing dihydropyridine receptors (DHPR) on the T-tubule membrane are activated by action potentials, promoting allosteric interaction with the voltage-gated ryanodine receptors (RyR) located on the proximal SR membrane to allow release of Ca^2+^ from the SR lumen. Transient increase in the concentration of cytoplasmic Ca^2+^ eliminates the inhibitive effect of troponin and tropomyosin on the actin-myosin interaction, thus activating the contractile apparatus, and a state of relaxation is restored when the cytoplasmic Ca^2+^ is re-uptaken into the SR through the SR Ca^2+^ adenosine triphosphatase (SERCA) pump [53,54,55,56]. E-C coupling in cardiac muscle is mediated by homologous proteins, though the DHPR complex and RyR1 are replaced by a different L-type voltage sensing channel and RyR2, respectively [57].

Murine models of SPEG deficiency have demonstrated substantial defects in E-C coupling and calcium handling in both cardiac and skeletal muscle [14,15]. Mice with adult-onset SPEG deficiency demonstrate HF preceded by T-tubule disruption [14], indicating that SPEG’s role in the organization of E-C coupling machinery is not limited to the developmental period. Huntoon et al. [15] further demonstrated that *Speg*-knockout (KO) skeletal muscles exhibited a marked decrease in the structural integrity and number of triads relative to controls correlating with severe deficits in muscle contractility and force and calcium mishandling. Both studies indicate that structural and functional defects of the E-C machinery are at the core of the SPEG-deficient phenotype, although the molecular mechanisms by which SPEG regulates E-C coupling are poorly understood, leaving a large gap in our understanding of the processes underlying muscle function and development.

Muscle development is a complex and highly coordinated process that occurs in distinct sequential stages (reviewed by Chal et al. [58]). Primary myogenesis occurs in the embryonic stage, wherein somitic stem cells give rise to Pax7^+^ myogenic progenitors, or myoblasts [59], which proliferate rapidly in the presence of fibroblast growth factor [60,61]. During the perinatal and neonatal stages, the vast majority of these myoblasts terminally differentiate into cardiomyocytes or fuse into multinucleated myotubes, which subsequently mature into skeletal myofibers. Remaining myoblasts become the source of adult myogenic stem cells, or satellite cells (SCs), localizing to specialized niches on mature myofibers [62,63]. Satellite cells remain in quiescence unless activated by muscle injury or atrophy signaling pathways, at which point they migrate to the infarcted site and proliferate to promote tissue regeneration and repair [64,65,66,67,68,69,70,71,72].

SPEG plays a significant role in muscle development and regeneration. SPEG deficiency in both cardiac and skeletal muscle has been associated with abnormal size, disorganization, and degeneration of myofibrils, central nucleation, sarcomeric fragmentation, and irregular or non-striated appearance of muscle tissue [8,15,16,36,73]. In addition to structural defects, cardiac progenitor cells (CPCs) from SPEG-deficient mice show defects in clone formation, growth, and differentiation into cardiomyocytes in vitro [73], which are associated with cardiac dysfunction in vivo. Further, administration of wild-type (WT) CPCs into the fetal hearts of SPEG-deficient mice has been shown to promote CPC engraftment and differentiation and enhance myocardial maturation, which rescues these mice from neonatal heart failure. Our recent study using striated muscle-specific *Speg*-KO mice shows that SPEG-deficient skeletal muscles contained fewer myogenic cells, which demonstrated reduced proliferation and delayed differentiation compared with those from WT muscles [20]. Regenerative response to skeletal muscle injury revealed similar abnormalities in *Speg*-KO mice, including fewer SCs, fewer dividing cells, and an increase in apoptotic cells. Future research will be required to characterize the process through which SPEG regulates CPC or SC differentiation and function during muscle development and regeneration.

## 6. SPEG-Related Pathways in E-C Coupling

E-C coupling and triadic structural defects are commonly observed in the various genetic forms of CNMs, indicating that genes linked to this disease likely engage in a common pathway that is integral to the formation and function of the E-C coupling machinery [74,75,76,77,78,79,80,81]. Recent studies have begun to elucidate and synthesize isolated fragments of the triad regulatory interactome, highlighting numerous pathways through which SPEG likely affects E-C coupling. We have identified MTM1, a lipid phosphatase responsible for regulating phosphoinositide levels at the sarcolemma, as a novel binding partner of SPEG in skeletal muscle [8]. MTM1 is believed to regulate E-C coupling by anchoring and enhancing the activity of BIN1 at the plasma membrane [82], where BIN1 works antagonistically with the large GTPase DNM2 to promote T-tubule formation [4]. Additionally, the catalytic domain of MTM1 has been shown to bind directly with the BAR domain (and, to a lesser extent, the SH3 domain) of BIN1 in a manner that is dependent on the conformation of BIN1 [82]. A perfect balance between the tubulation activity of BIN1 and the fission activity of DNM2 is required for proper T-tubule development, and disruptions in this signaling pathway could lead to severe triadic structural defects. Of note, each gene involved in this pathway has been linked to a unique form of CNM [3,4,83], indicating the presence of a master underlying mechanism centered around the triadic regulatory pathway. Furthermore, several previous studies have demonstrated that DNM2 is consistently upregulated in the phenotypes of *MTM1*-, *BIN*1-, and *DNM2*-linked CNM [12,84,85,86,87,88,89], providing a direct explanation for the T-tubule disruptions that have been reported [14,15,74,77,81,90]. Further investigation will be required to experimentally verify and characterize this proposed pathway, and to elucidate the specific role of SPEG within it.

In addition, several additional pathways (reviewed by Campbell et al. [23]) through which SPEG may regulate cardiac calcium homeostasis have been identified (Figure 2) [14,17,19,21,36,91]. In SPEG-deficient cardiac tissue, the disruption of T-tubule and junctional membrane complex (JMC) occurs prior to HF onset and was possibly attributed to reduced SPEG-regulated phosphorylation of junctophilin-2 (JPH2) [14]. Cardiac-specific *Speg*-KO mice also exhibit increased Ca^2+^ spark frequency indicative of aberrant RyR2 activity, as well as decreased Ca^2+^-transient amplitudes indicative of reduced systolic SR Ca^2+^ release along with reduced SR Ca^2+^ load [14]. In addition to affecting sarcomere and JMC structure, SPEG can directly affect E-C coupling by phosphorylating key Ca^2+^-handling proteins. For example, SPEG is capable of binding to RyR2 using the N-terminal portion of SPEG [14], that is unique to SPEGβ, and phosphorylating RyR2 at residue S2367 to reduce its activity [21]. Similarly, loss of function of Unc-89, an obscurin-MLCK found in Drosophila and Caenorhabditis elegans, led to RyR dysregulation and contractile dysfunction in *C. elegans* [25,92]. SPEG can also phosphorylate SERCA2a through its SK-2 domain, which enhances SR Ca^2+^-reuptake activity [19]. In skeletal muscle, SPEG deficiency was associated with a severe depression of the Ca^2+^ current function of the voltage sensor of E–C coupling, and a profoundly affected SR Ca^2+^ release [15]. It remains to be investigated whether SPEG in the skeletal muscle regulates E-C coupling in a similar or tissue-specific manner.

## 7. SPEG and Its Interacting Partners

### 7.1. Ig Like and FnIII Domains

The Ig-like and FnIII domains facilitate protein-protein interactions that allow MLCK family proteins to help form a cytoskeletal network and compartmentalize the cell [93]. In skeletal muscle, we have demonstrated that SPEG interacts with Ig-like/FnIII of MTM1 and desmin [8,22] (Table 1). These two proteins are critical to the integrity and function of the muscle triads.

MTM1, associated with the X-linked form of centronuclear myopathy, is a lipid phosphatase that is involved in the biogenesis and maintenance of membrane homeostasis and muscle structure. It is responsible for regulating phosphoinositide levels at the membrane, where it uses phosphatidylinositol 3-phosphate and phosphatidylinositol-3,5-diphosphate to generate the products phosphatidylinositol and phosphatidylinositol 5-phosphate [94,95]. Our previous work revealed that the Ig-like domain 9 (amino acids 2583-2673) of SPEG interacts with both phosphatase and coiled-coil domains (amino acids 155-603) of MTM1 [8]. Subfragments including either the phosphatase or the coiled-coil domains alone did not interact with SPEG, suggesting that the critical binding region either overlaps the junction or might include cooperative binding sites in both regions of MTM1 [8].

Desmin represents the major cytoplasmic intermediate filaments (IF) in the muscle [96,97]. Desmin is the classical type III IF protein with a tripartite structure comprising a rod domain flanked by non-helical head and tail domains. The rod domain, formed by four-helical segments, is involved in protein–protein interactions and plays a critical role in desmin filament assembly and the formation of the extrasarcomeric cytoskeleton [98]. Mutations in desmin are associated with myopathy and cardiomyopathy [96,97]. We recently identified that Ig-like and FnIII domains (amino acids 2200-2960) of SPEG interact with the rod domain of desmin (amino acids 179−228). Similarly, neither Ig-like nor FnIII domain of SPEG alone showed interaction with desmin [22].

It has also been shown that MTM1 controls desmin IF architecture and mitochondrial dynamics in skeletal muscles and that deficiency of MTM1 causes increase in desmin expression and insolubility [96]. A relationship between SPEG and desmin proteins is suggested by the tandem arrangement of desmin- and SPEG-encoding genes within 8.3 kb of each other under the control of a common locus control region, as it shows an evolutionary pressure to coordinately regulate these proteins [40]. SPEG depletion also leads to desmin aggregates in vivo and a shift in desmin equilibrium from soluble to insoluble fraction [22]. Further, MTM1 and SPEG are two proteins altered in CNM, while recessive desmin-null muscular dystrophy present with increased central nucleation and mitochondrial abnormalities [99]. It is possible that the relationship between desmin and SPEG may be indirect with MTM1 serving as the common link, which deserves future investigation.

### 7.2. Kinase Domains

Conservation of the tandem kinase domains among Unc89 family proteins and across species indicates that these catalytic regions are critical to the proteins’ function. SPEG, unlike its Unc89 family members, expresses no isoforms that lack this dual kinase motif, further emphasizing the critical nature of these domains. Thus far, four targets of SPEG kinase activity have been experimentally verified in cardiac muscle, JPH2, RyR2, SERCA2a, and tropomyosins [14,19,21,36] (Table 1).

JPH2 is the cardiac isoform of the junctophilins family that stabilizes the cardiac dyads between T-tubule and junctional SR membranes, which is critical for proper intracellular calcium signaling [100]. JPH2 knockdown leads to T-tubule disruption in developing myocardium and in the adult mouse heart [101,102]. By utilizing an unbiased proteomic analysis of immunoprecipitated JPH2 and RyR2 from adult mouse hearts, Quick et al. first identified SPEG as the only binding partner for both proteins [14]. Further in vitro co-expression and immunoprecipitation studies revealed that JPH2 binds SPEGα whereas RyR2 binds the N-terminal region of SPEGβ. Loss of SPEG in mouse hearts causes JPH2 dephosphorylation, although the phosphorylation levels of RyR2 at the sites of S2808 and S2814, which were known to contribute to Ca^2+^ dysregulation in heart failure, were unaltered. Quan et al. further showed that the SK1 of SPEG interacts with and phosphorylates JPH2 [19]. However, the exact residue(s) on JPH2 that are subject to SPEG phosphorylation remain to be determined.

RyR2, anchored to the SR, is the major mediator for the sarcoplasmic release of stored calcium ions [103]. Campbell et al. recently identified that SPEG phosphorylates RyR2 at a previously uncharacterized serine (S2367) located in the central domain of the channel., and that loss of SPEG causes reduced S2367 phosphorylation on RyR2 [21]. In contrast to previously characterized phosphorylation sites (S2030, S2808, and S2814) on RyR2, S2367 phosphorylation inhibits diastolic Ca^2+^-release from RyR2 while loss of phosphorylation of this site increases atrial fibrillation susceptibility. The identity of the kinase in SPEG responsible for phosphorylating RyR2 remains unknown [14]. It is speculated that RyR2 is modified by the SK1 of SPEG since RyR2 can be modified by the first kinase domain of obscurin which is very similar to the SK-1 of SPEG [104].

SERCA2a, mainly expressed in cardiomyocytes, mediates calcium reuptake from the cytoplasm into the SR, maintaining calcium homeostasis [105]. Quan et al. reported that SPEG also interacts with SERCA2a and that the SK2 of SPEG phosphorylates Thr^484^ on SERCA2a [19]. Inducible deletion of *Speg* decreased SERCA2a-Thr^484^ phosphorylation and its oligomerization in the heart, which further inhibited its calcium-transporting activity and impaired calcium reuptake into the SR in cardiomyocytes. They further characterized that SPEG is phosphorylated on Ser2461/Ser2462/Thr2463 by protein kinase B (PKB) in response to insulin and that PKB-mediated phosphorylation of SPEG activates its second kinase-domain, which in turn phosphorylates SERCA2a and accelerates calcium re-uptake into the SR [17].

α-tropomyosin, which binds with actin in muscle to form the backbone of the thin filament and works in conjuction with the troponin complex to regulate the contractile apparatus is also phosphorylated by SPEG in cardiac tissue [36,106,107] providing a direct link between Speg activity and development of the sarcomeric architecture.

The kinase activity of SPEG has not yet been evaluated in skeletal muscle, although preliminary yeast two-hybrid screening of SPEG’s second kinase domains revealed interactions with junctophilin-1 (JPH1), which functions analogously in skeletal muscle to JPH2 in heart [108], and multiple tropomyosins. The functional consequences of these relationships in skeletal muscle will need to be experimentally evaluated.

## 8. Conclusions

SPEG mutations are associated with CNM, DCM, or a combination of both CNM and DCM. Mutations that only affect the region unique to SPEGβ with preserved SPEGα function appear to be associated with a milder cardiomyopathy phenotype. Similarily, mutations involving the two kinase domains seem to be predominantly associated with DCM indicating that SPEG may have tissue-specific roles unique to its isoforms or functional domains. Disease modelling of SPEG alterations in mice and iPS cells has provided with insights into the mechanisms by which SPEG alterations could cause CNM and DCM. SPEG has several functions in skeletal and cardiac muscles, including CPC/SC differentiation and muscle regeneration, stabilization of T-tubules and E-C coupling machinery, regulation of SR Ca^2+^ release via RyR2, and modulation of SR Ca^2+^ reuptake via SERCA2a. Given SPEG’s localization to the sarcolemma as well as our recent finding of focal adhesion defects after SPEG deletion, it will be important to examine whether SPEG has additional targets in these regions or potential structural functions. Recent research has also begun to elucidate a functional network in which MTM1, BIN1, and DNM2 interact with one another to regulate triad formation through a mechanism of membrane trafficking and remodeling. We have previously demonstrated that SPEG interacts and co-localizes with MTM1. It is critical to elucidate how SPEG fits into this functional network which may provide with clues into treating SPEG-related diseases in the future. 

## Figures and Tables

**Figure 1 ijms-22-05732-f001:**
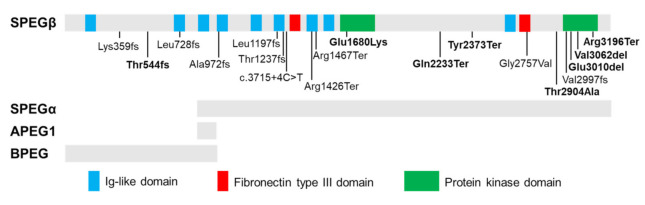
Tissue-specific isoforms of SPEG and disease-related recessive mutations. Schematic of full-length SPEG protein showing its functional domains, including nine immunoglobulin (Ig)-like, two fibronectin Type III and two protein kinase domains, with positions of identified mutations (homozygous mutations: in bold). The gray bars below the diagram indicate which domains are part of the four SPEG isoforms (SPEGβ, SPEGα, APEG1, and BPEG) as marked below. Spegα and Spegβ are predominantly expressed in skeletal and cardiac muscle; APEG1 is expressed predominantly in vascular smooth muscle cells (VSMCs); BPEG is expressed exclusively in brain and aorta.

**Figure 2 ijms-22-05732-f002:**
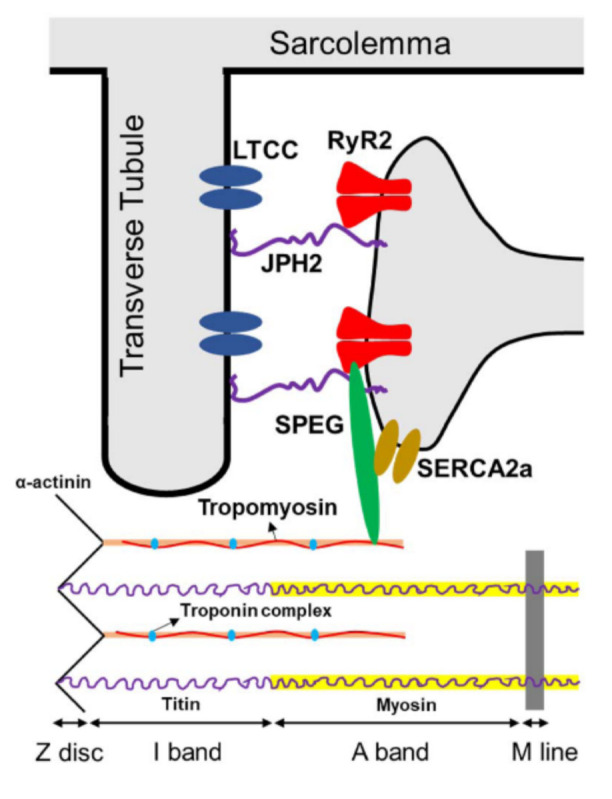
Known targets of SPEG phosphorylation in cardiac muscles, including JPH2, RyR2, SERCA2a, and α-tropomyosin. Abbreviations: SPEG = striated preferentially expressed protein kinase; JPH2 = junctophilin 2; LTCC = L-type calcium channel; RyR2 = ryanodine receptor 2; SERCA2a = sarco-endoplasmic reticulum ATPase-2a.

**Table 1 ijms-22-05732-t001:** SPEG and its interacting partners in skeletal and cardiac muscle.

SPEG Domain	Interacting Protein	Domain or Phosphorylation Site (PS) on Interacting Protein	Biological Significance	Evidence from	Refs
Ig like and FnIII domain	Myotubularin (MTM1)	Phosphatase and coiled-coil domains (amino acids 155-603)	To be determined	Skeletal muscle	[8]
	Desmin (DES)	Rod domain (amino acids 179−228).	To be determined		[22]
Kinase domain	Junctophilin 2 (JPH2)	To be determined	Transverse tubule formation and maintenance	Cardiac muscle	[14,19]
	Ryanodine receptor (RyR2)	Ser2367 (PS)	Inhibition of RyR2 and diastolic Ca^2+^ release		[14,21]
	Sarcoplasmic/endoplasmic reticulum Ca^2+^ ATPase 2a (SERCA2a)	Thr484 (PS)	Increase of SERCA2a oligomerization and calcium reuptake into SR		[19]
	α-tropomyosin (TPM1)	To be determined	To be determined		[36]

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
