# Peer review of "Striated Preferentially Expressed Protein Kinase (SPEG) in Muscle Development, Function, and Disease"

_ijms, 2021, doi:10.3390/ijms22115732_

Round 1

Reviewer 1 Report

This paper is dedicated to  the actual research problem  - CNMs, congenital myopathies. Process of  SPEG gene encoding to MTM1, DNM2, and BIN1 highly similar pathogenesis  is clearly described . Degeneration  of striated muscle tissue , fragmentation sarcomeres ,  role of satellite cells  in myogenesis, mechanisms of this highly coordinated process is  well described. Paper is actual from theoretical as well as from clinical aspects. My suggestion - publish in present form.

Author Response

Thanks for the comments.

Reviewer 2 Report

This is a well-written and well-reasoned review focusing on the role of Striated preferentially expressed protein kinase in muscle development, function, and disease. 

  1. The sentence on lines 147-149 is unclear, and citing figure 1 seems to be out of context.
  2. A visual or graphical depiction and tables in a review article are more appealing than the text portion. The authors should include a few figures and tables to improve the quality of the paper.
  3. "Figure 1. Tissue-specific isoforms of SPEG and disease-related recessive mutations." here readers can not understand the specificity of tissue by this figure
  4. "SPEG and its interacting partners" a  brief listing in a tabular form will be more attractive
  5. Concluding remarks should be compact and generally without references
  6. There is no good connection between the abstract and the conclusion.

Author Response

  1. The sentence on lines 147-149 is unclear, and citing figure 1 seems to be out of context.

Response: We have revised this sentence. Citing figure 1 is to show the location and mutation type of all reported SPEG variants.

  1. A visual or graphical depiction and tables in a review article are more appealing than the text portion. The authors should include a few figures and tables to improve the quality of the paper.

Response: Thanks for the comments. We have added Table 1 to summarize the interacting partners of SPEG for a better view. Although much progress has been made in understanding the function of SPEG kinase in cardiac muscles (as shown in figure 2), its role in skeletal muscle and the disease mechanism underlying the triad abnormalities and satellite cell defects after SPEG depletion remains to be elucidated.   

  1. "Figure 1. Tissue-specific isoforms of SPEG and disease-related recessive mutations." here readers cannot understand the specificity of tissue by this figure.

Response: Thanks for the comments. We have explained the tissue-specific isoforms in the figure caption for better understanding.

  1. "SPEG and its interacting partners" a brief listing in a tabular form will be more attractive.

Response: We have added Table 1 to summarize the interacting partners of SPEG.

  1. Concluding remarks should be compact and generally without references.

Response: Thanks for the comments. We have revised this part and removed the references.

6. There is no good connection between the abstract and the conclusion.
Response: We have revised the abstract for a better summary of the context.